# Trajectory Tracking Control of Variable Sweep Aircraft Based on Reinforcement Learning

**DOI:** 10.3390/biomimetics9050263

**Published:** 2024-04-27

**Authors:** Rui Cao, Kelin Lu

**Affiliations:** 1The College of Information Engineering and Artificial Intelligence, YangZhou University, Yangzhou 225009, China; stdio@yzu.edu.cn; 2School of Automation, Southeast University, Nanjing 210096, China

**Keywords:** morphing aircraft, deep deterministic policy gradient, path tracking, environmental disturbance

## Abstract

An incremental deep deterministic policy gradient (IDDPG) algorithm is devised for the trajectory tracking control of a four-wing variable sweep (FWVS) aircraft with uncertainty. The IDDPG algorithm employs the line-of-sight (LOS) method for path tracking, formulates a reward function based on position and attitude errors, and integrates long short-term memory (LSTM) units into IDDPG algorithm to enhance its adaptability to environmental changes during flight. Finally, environmental disturbance factors are introduced in simulation to validate the designed controller’s ability to track climbing trajectories of morphing aircraft in the presence of uncertainty.

## 1. Introduction

The morphing aircraft is a novel conceptual aircraft with deformable structure, which can autonomously adapt its morphology in response to changes in flight environment and mission requirements [1,2]. In comparison to traditional fixed-wing aircraft, morphing aircraft exhibit significant advantages on multiple fronts: firstly, the deformable structure of morphing aircraft can be utilized to enhance aerodynamic characteristics, reduce flight energy consumption, and expand the flight envelope; secondly, active morphing assists in maneuvering, thereby augmenting control capabilities; additionally, morphing aircraft can adapt to various flight environments and mission requirements by altering configurations, thereby broadening its applicability [3,4].

For morphing aircraft, each configuration will generate unique performance (such as speed, climb rate, range, endurance, etc.) under specific flight conditions (Mach number, altitude, angle of attack, and sideslip angle). For a given flight profile, optimal morphologies for various flight phases can typically be theoretically calculated. However, in practical flight missions, obtaining all optimal configurations within the entire flight envelope is challenging. Moreover, mission parameters may be modified or entirely changed during flight. In such cases, the deformation strategy derived from the theoretical method may not be globally optimal. Real-world mission requirements underscore the need for morphing aircraft to evolve toward intelligence and autonomy to effectively adapt to increasingly complex flight environments.

In the past three decades, artificial intelligence (AI) technologies have undergone rapid development [5,6]. Various AI approaches, such as reinforcement learning [7,8] and deep reinforcement learning [9], have provided more intelligent solutions for deformation strategies in morphing aircraft. Deep learning, a key branch of machine learning, employs multi-layer neural networks for data perception and representation, demonstrating robust capabilities in handling complex classification tasks [10]. Reinforcement learning engages in iterative learning through continual trial and error, enhancing its ability to make behavioral decisions by interacting with the environment and receiving feedback. The core of reinforcement learning lies in modifying its own strategy based on evaluative feedback signals from the environment, thereby achieving decision optimization [11]. Common reinforcement learning algorithms include deep Q-network (DQN) [12], deep deterministic policy gradient (DDPG) [8], proximal policy optimization (PPO) [13], among others. Ref. [14] proposes a Q-learning-based adaptive control method for a swept-wing aircraft. However, this method faces limitations due to an excessively discrete action space. Additionally, both the Q-function and reward function are solely dependent on the aircraft’s configuration rather than its flight state, resulting in a narrow applicability scope. The DDPG algorithm is a commonly used algorithm in deep reinforcement learning, combining deterministic policy gradient methods with DQN. It employs deep neural networks to approximate state action policies and is suitable for handling continuous action problems. Ref. [15] designs a DDPG algorithm to learn deformation strategies under both symmetric and asymmetric conditions for a morphing aircraft with simultaneous changes in wing span and sweep angle. The literature [16] introduces an enhanced DDPG algorithm for the design of deformation strategies in a morphing unmanned aerial vehicle (UAV), allowing for the deformation of wing sweep angle, wing span, wing area, and other flight structures based on environmental conditions and mission objectives.

Reinforcement learning control strategies enable aircraft to converge from various initial states to predefined endpoints. However, during the training of the learning algorithm, the large action space often results in slow training speeds. To address this issue, this study draws inspiration from trajectory tracking control design principles. In the simulated environment of reinforcement learning, a reference trajectory is introduced. Thus, the action space is defined as the error between the current action and the reference trajectory, thereby reducing the action search space. Additionally, system reference trajectories are typically based on nominal models, but actual systems may exhibit model uncertainties and obstacles [17,18]. Model uncertainty significantly affects the control effectiveness of traditional trajectory tracking control methods. Addressing the challenges posed by model uncertainty and obstacle presence, this study incorporates long short-term memory (LSTM) recursive neural networks [19,20] into the reinforcement learning algorithm to record the positions of reference trajectories and obstacles. Consequently, reinforcement learning-based trajectory tracking control can optimize an optimal trajectory and control strategy near the reference trajectory, matching the real model, and enhancing computational efficiency based on historical flight data.

This paper introduces an incremental model with uncertainties for a four-wing variable-sweep (FWVS) aircraft and redesigns the action and state spaces based on this model. In Section 3, line-of-sight (LOS) is employed as a path-tracking method with a reference trajectory, and a reward function based on position and attitude errors is designed. Section 4 integrates the LSTM into the DDPG algorithm framework to enhance its real-time adaptability to environmental changes during flight. The resulting algorithm, IDDPG, is established. In Section 5, through simulation, the IDDPG algorithm demonstrates faster convergence and achieves integrated tracking control of climb trajectories for a variant aircraft model with uncertainties using deep reinforcement learning. To assess the adaptability of the controller to environmental disturbances, such as obstacles, simulations with environmental interference are conducted. Section 6 provides a summary of the entire work.

## 2. Model Description

### 2.1. Mathematical Model of Four-Wing Variable-Sweep Aircraft

The subject of this study is a FWVS aircraft with a tandem wing configuration. Typically, this type of tandem wing aircraft employs a passive variable-sweep angle scheme, with the main wings folding before takeoff and expanding during flight to reduce the fuselage size while ensuring sufficient lift for the aircraft [16]. This type of aircraft can control flight attitude and trajectory by adjusting the sweep angle of its four main wings, simplifying control surfaces. In principle, this design enables a stable flight without ailerons or horizontal tails, presenting significant practical and theoretical research value. It is important to note that this morphing aircraft only considers symmetric deformation.

The control inputs for the FWVS aircraft are the variation in the sweep angles of its four main wings, as shown in Figure 1. These angles represent the rotation of the *z*-axis in each wing coordinate system with respect to the body coordinate system. Here, the $x_{i}$-axis is aligned with the wing coordinate system’s $Ox_{a}y_{a}$ plane, and the $y_{i}$-axis passes through the wing rotation center, pointing to the right side of the aircraft. The sweep angles of each main wing are denoted as $a_{1},a_{2},a_{3},a_{4}$.

Adopting clockwise rotation as positive, during symmetric deformation, $a_{1}=-a_{2}<0,$ $a_{3}\;=-a_{4}<0$, and $\left|a_{i}\right|\in \left[0^{o},30^{o}\right]$. Following the definition in Ref. [21], the control variable is simplified as the deformation rate:(1)$$\begin{array}{c} \lambda_{1}=\frac{\left|a_{1}\right|}{30^{o}},\lambda_{2}=\frac{\left|a_{3}\right|}{30^{o}}, \end{array}$$
that is $\lambda_{i}\in \left[0,1\right]$. i=1,2 represent the front wing and rear wing, respectively.

This study employs OPENVSP 3.21.1 software for aerodynamic analysis of the FWVS aircraft. Under various sweep angle deformations and flight conditions (including angle of attack, Mach number, and Reynolds number), the longitudinal aerodynamic force and moment coefficients of this aircraft are computed. Ultimately, polynomial fitting is utilized to obtain expressions for lift coefficient $C_{L}$, drag coefficient $C_{D}$, and pitch moment coefficient $C_{m}$.
(2)$$\begin{array}{cc} C_{L}=\; & 0.01\times \left(47.95-4.077\lambda_{1}^{2}-4.579\lambda_{2}^{2}+16.89\lambda_{1}^{2}\lambda_{2}\right.\\ \; & \left.+17.44\lambda_{1}\lambda_{2}^{2}-20.41\lambda_{1}\lambda_{2}-16.23\lambda_{1}^{2}\lambda_{2}^{2}\right)\left(9.448\alpha +0.3397\right),\\ C_{D}=\; & 0.01\times \left(83.58-5.229\lambda_{2}-0.1296\lambda_{2}^{2}-4.34\lambda_{1}\right.\\ \; & \left.-0.409\lambda_{2}+3.595\lambda_{1}\lambda_{2}\right)\left(28.85\alpha^{2}+0.2363\alpha +0.8429\right),\\ C_{m}=\; & 0.01\times \left(-8.103-11.67\lambda_{1}^{2}-4.252\lambda_{2}^{2}-27.26\lambda_{1}+37.47\lambda_{2}\right)\times \\ \; & \left(-8.207\alpha^{2}+10.03\alpha +0.34\right)+\left(7.248\lambda_{1}+36.89\lambda_{2}-69.24\right)q, \end{array}$$
where α is the angle of attack, and *q* denotes the pitch angle rate.

During the deformation process, changes in the sweep angle induce variations in the center of mass and aerodynamic center, leading to alterations in pitch moment. The rotational velocity and acceleration of the wing give rise to additional inertial forces $F_{xdyn}$, $F_{zdyn}$, and moment $M_{dyn}$, which can be expressed as follows [21]:(3)$$\begin{array}{c} \left\{\begin{array}{c} F_{z}=mg\cos\theta -D\sin\alpha -L\cos\alpha \\ F_{xdyn}=2m_{a}l\left({\dot{a}}_{1}^{2}\cos a_{1}-{\ddot{a}}_{1}\cos a_{1}+{\dot{a}}_{3}^{2}\cos a_{3}-{\ddot{a}}_{3}\cos a_{3}\right)\\ F_{zdyn}=2m_{a}l\left({\dot{a}}_{1}^{2}\sin a_{1}-{\ddot{a}}_{1}\sin a_{1}+{\dot{a}}_{3}^{2}\sin a_{3}-{\ddot{a}}_{3}\sin a_{3}\right)\\ J_{y}=0.0242+4\times 2.6\times 10^{-5}+4m_{a}c^{2}+m_{a}\left(l_{1}^{2}+l_{2}^{2}+l_{3}^{2}+l_{4}^{2}\right)\\ M_{dyn}=2m_{a}ql\left({\dot{a}}_{3}l_{2}\cos a_{3}-{\dot{a}}_{1}l_{1}\cos a_{1}\right)+\\ \;\;\;\;\;\;\;\;\;\;\;\;\;2m_{a}l\left(q\sin\theta +Fz/m\right)\left(\sin a_{1}+\sin a_{3}\right)-\\ \;\;\;\;\;\;\;\;\;\;\;\;\;2m_{a}cl\left({\dot{a}}_{1}\cos a_{1}-{\ddot{a}}_{3}\cos a_{3}-{\dot{a}}_{1}^{2}\sin a_{1}+{\dot{a}}_{3}^{2}\sin a_{3}\right)\\ M_{Gdyn}=-2m_{a}gl\cos\theta \left(\sin a_{1}+\sin a_{3}\right) \end{array}\right. \end{array}$$
where $F_{z}$ represents the sum of external forces acting in the body’s *z*-axis direction, $J_{y}$ denotes the moment of inertia, *c* is the mean aerodynamic chord length, *l* is the mean arm length, $l_{i}$ is the wingspan of the *i*th main wing, and $M_{Gdyn}$ represents the additional inertial pitch moment caused by changes in the center of mass.

This paper primarily focuses on the longitudinal motion of the FWVS aircraft. Therefore, it is assumed that the lateral attitudes are all zero. Consequently, the simplified longitudinal kinematic equations can be obtained as follows:(4)$$\begin{array}{c} \left\{\begin{array}{c} \dot{x}=V\cos\gamma \\ \dot{h}=V\sin\gamma \\ \dot{V}=\left(T\cos\alpha -D-mg\sin\gamma +F_{xdyn}\cos\alpha +F_{zdyn}\sin\alpha \right)/m\\ \dot{\gamma }=\left(T\sin\alpha +L-mg\cos\gamma -F_{zdyn}\cos\alpha +F_{xdyn}\sin\alpha \right)/mV\\ \dot{\alpha }=q-\dot{\gamma }\\ \dot{q}=\left(M+M_{Gdyn}+M_{dyn}\right) \end{array}\right. \end{array}$$
where *T* is thrust, *x* denotes horizontal displacement, *h* represents flight altitude, *V* indicates flight velocity, and γ expresses the flight path angle.

### 2.2. Kinematic Incremental Model of the FWVS Aircraft

Here, we first formalize the dynamic model of the FWVS aircraft into a general form. The aircraft controls its position and attitude by simultaneously adjusting the sweep angles of the front and rear wings, as well as the thrust magnitude. Thus, in Equation (Equation 4), $u=\left[\lambda_{1},\lambda_{2},T\right]$ represents the control input and $X=\left[x,h,V,\alpha ,\gamma ,q\right]$ expresses the output state variables, and we can obtain
(5)$$\begin{array}{c} \dot{X}=f\left(X,u\right). \end{array}$$

Based on the description of the aircraft kinematics in Equation (Equation 5), for a given reference trajectory $X_{r}=\left[x_{r},h_{r},V_{r},\alpha_{r},\gamma_{r},q_{r}\right]$ and $u_{r}=\left[{\lambda_{1}}_{r},{\lambda_{2}}_{r},T_{r}\right]$, the following representation can be formulated:(6)$$\begin{array}{c} \dot{X_{r}}=f\left(X_{r},u_{r}\right), \end{array}$$
among them, *r* represents the reference state.

By subtracting Equation (Equation 5) from Equation (Equation 6), the incremental model for this morphing aircraft can be obtained
(7)$$\begin{array}{c} \Delta \dot{X}=f\left(\Delta X+X_{r},\Delta u+u_{r}\right)-f\left(X_{r},u_{r}\right), \end{array}$$
where $\Delta X=X-X_{r},\Delta u=u-u_{r}$.

Due to the strong nonlinearity and time-varying characteristics of this morphing aircraft, obtaining an accurate mathematical model is challenging. Reinforcement learning is particularly suitable for such cases with imprecise models. Considering the uncertainty in the model, this paper incorporates it into the model parameters. We assume uncertainty in the lift coefficient of the system, expressed in the following form:(8)$$\begin{array}{c} \begin{array}{cc} C_{L} & =\left(0.01\times \left(47.95-4.077\lambda_{1}^{2}-4.579\lambda_{2}^{2}+16.89\lambda_{1}^{2}\lambda_{2}\right.\right.\\ \; & \left.\left.+17.44\lambda_{1}\lambda_{2}^{2}-20.41\lambda_{1}\lambda_{2}-16.23\lambda_{1}^{2}\lambda_{2}^{2}\right)\left(9.448\alpha +0.3397\right)\right)\left(1+\varepsilon \right) \end{array} \end{array}$$
where $\varepsilon \in \left[-0.2,+0.2\right]$ represents an uncertain parameter with fixed boundaries. This indicates that the range of uncertainty for $C_{L}$ is between −20% and 20%. Finally, the lift coefficient $C_{L}$ is incorporated into the incremental model (Equation 7) to obtain the incremental model of the FWVS aircraft with uncertainty.

## 3. Reinforcement Learning Environment Design

For reinforcement learning, if the action space is designed to be excessively large, it can result in high computational complexity during network training and slow convergence speed. Considering that an aircraft follows a corresponding reference trajectory during actual flight, to address the issue of slow training convergence due to an excessively large action space, this study constrains the action space around the reference trajectory during network training. This approach aims to reduce computational complexity. In configuring the reinforcement learning environment, the incremental kinematic model based on the reference trajectory, as depicted in Equation (Equation 7), is employed.

### 3.1. Action Space and State Space Design

To apply reinforcement learning algorithms to the FWVS aircraft, it is essential to establish the reinforcement learning environment. It includes defining and designing the state space, action space, and reward function. The reinforcement learning environment not only needs to be a Markov decision process but also must meet the mission requirements of the aircraft. In this section, the deformation process is represented as a standard Markov decision process [22], consisting of states *S*, actions *A*, rewards *R*, and discount factor η. If we analogize the reinforcement learning model to a traditional controller, the inputs can be likened to actions and outputs can be considered as state values. Hence, the state variables consist of current flight parameters describing the flight status of the aircraft, specifically flight altitude *h*, horizontal displacement *x*, flight speed *V*, pitch angle rate *q*, angle of attack α, and flight path angle γ.

The reinforcement learning task designed in this study is trajectory tracking. To reduce exploration in the action space and expedite convergence, the action space is designed in proximity to the reference trajectory. Specifically, it is represented as $A=\left[\delta_{\lambda 1},\delta_{\lambda 2}\right]\in \left[-0.1,0.1\right]$, signifying the selection constraint of actions within a range of ±0.1 from the reference trajectory. Here, $\delta_{\lambda_{i}}=\lambda_{i}-\lambda_{ri}$ represents the difference between the current actual value and the reference value. Based on the flight mission and reference trajectory data for the variant aircraft, a fixed thrust value of 7.3 N is set during the actual algorithm execution.

### 3.2. Reward Function Design Based on LOS

Reinforcement learning is an algorithm that relies on reward functions for training, analogous to deep learning where the reward function serves as a supervisory signal. Therefore, a reasonable designed reward function is crucial. In the field of reinforcement learning, reward functions can be categorized as sparse and non-sparse rewards [23]. Sparse rewards imply that an effective reward signal is received only after completing a specific task. This implies that, during the interaction between the agent and the environment, no rewards are obtained, which is highly detrimental to exploration. Consequently, the agent receives a reward of zero for most actions due to the vast action space, leading to numerous futile explorations and slow convergence. Hence, this paper adopts a composite reward function combining sparse rewards with non-sparse rewards to expedite convergence while actively guiding the completion of flight tasks.

#### 3.2.1. Sparse Reward Design

Initially, a sparse reward approach is employed to address the task success, designing a terminal reward. Physically, upon reaching the target point, the agent receives a positive reward, while exceeding a threshold based on the aircraft’s own state results in a negative reward. If either condition is triggered, the state of the variant aircraft is reset, initiating the next flight task. The design is as follows:

Initially, a sparse reward mechanism is employed to design the terminal reward. Physically, upon reaching the target point, the aircraft receives a positive reward $r_{c}$, while exceeding the state threshold of aircrafts results in a negative reward $r_{o}$. If either condition is triggered, the state of the variable aircraft is reset, initiating the next flight task. The design of $r_{c}$ is as follows:(9)$$\begin{array}{c} r_{c}=t\leq t_{f}\&\left|h-h_{\mathrm{target}}\right|\leq \delta_{h}\&\left|V-V_{\mathrm{target}}\right|\leq \delta_{V}\&\left|\gamma -\gamma_{\mathrm{target}}\right|\leq \delta_{\gamma }\&\left|q\right|\leq \delta_{q}, \end{array}$$
where $t_{f}$ represents the maximum time to complete the task. $h_{\mathrm{target}}$, $V_{\mathrm{target}}$, and $\gamma_{\mathrm{target}}$ denote the target values for flight altitude, speed, and flight path angle, respectively. $\delta_{h}$, $\delta_{V}$, $\delta_{\gamma }$, and $\delta_{q}$ express the maximum allowable error ranges for altitude, speed, flight path angle, and pitch angle rate, respectively.

The design of $r_{o}$ is as follows:(10)$$\begin{array}{c} r_{o}=t>t_{f}||h\notin \left[h_{\min},h_{\max}\right]||V\notin \left[V_{\min},V_{\max}\right]\left|\right|\left|q\right|>q_{\max}||\gamma \notin \left[\gamma_{\min},\gamma_{\max}\right]\left|\right|\left|\alpha \right|>\alpha_{\max}, \end{array}$$
where $h_{\max}$ and $h_{\min}$ represent the maximum and minimum values for flight altitude constraints, respectively. $V_{\max}$ and $V_{\min}$ denote the maximum and minimum values for flight speed constraints. $q_{\max}$ expresses the maximum permissible pitch angle rate and $\alpha_{\max}$ signifies the maximum allowable angle of attack. $\gamma_{\max}$ and $\gamma_{\min}$ are the maximum and minimum values for flight path angle constraints.

Furthermore, to enhance the reward for exemplary flight states and increase the diversity of samples [24], substantial coefficient factors are applied to the two rewards mentioned above. The terminal reward $R_{f}$ is then obtained by combining the two designed rewards:(11)$$\begin{array}{c} R_{f}=600r_{c}-100r_{o}. \end{array}$$

#### 3.2.2. Non-Sparse Reward Design

Non-sparse reward is a deliberately crafted "dense reward", typically manifested as a reward function associated with the state. To enhance the utility of flight process samples, the following action reward function and state reward function have been designed based on the concept of non-sparse rewards:

(1) Action Reward: The morphing aircraft must ensure stable flight during the deformation process, and to achieve smoother trajectories, an action penalty function $R_{a}$ is designed as follows:(12)$$\begin{array}{c} R_{a}=w_{3}\left|\Delta A\right|, \end{array}$$
where ΔA represents the change in actions and $w_{3}$ is a negative value used to guide action $\lambda_{1}$ and $\lambda_{2}$ toward slow variations.

(2) LOS-based State Reward: The reinforcement learning algorithm should guide the FWVS aircraft to track the reference trajectory and accomplish the tracking flight task. Therefore, a process reward $R_{m}$ based on the LOS method is designed.

The line-of-sight method [25] is a guidance approach that provides the controlled object with a visual range, aiming to perform path tracking within this range. For UAVs, the field of view is defined as a circular area with its own center of mass as the center and a radius of *R*. Assuming a decoupled vertical plane for path tracking, an appropriate radius is selected to ensure that this circle has two points with the target path. The closer intersection to the UAV is chosen as the target point. The direction from the aircraft’s current position to this point is considered as the target heading $\psi_{\mathrm{LoS}}$, where Δψ represents the angle between the heading line and the *x*-axis. Assuming that the heading direction and the desired path are in the same horizontal plane, the principle of this method is illustrated in Figure 2, when there are two intersections between the expected path and the circular field of view. Let $\left(x_{d},h_{d}\right)$ be the intersection point between the circular field of view and the desired path; then, the target heading angle can be obtained as follows:(13)$$\begin{array}{c} \Delta \psi =\arctan\frac{y_{d}}{\sqrt{R^{2}-y_{d}^{2}}}. \end{array}$$

Additionally, when there are no intersection points between the desired path and the circular field of view, i.e., when the UAV is too far from the desired path, the vertical direction is selected as the target heading. The objective is to approach the desired path at the fastest speed.

Subsequently, based on this method, a process reward for path tracking is designed. The variables directly related to the path include the relative distance *D* and heading angle ψ between the morphing aircraft and the reference trajectory, which can be obtained at any moment. Let the position vector at the current time be denoted as $X_{\mathrm{UAV}}=\left[x_{1},h_{1}\right]$, and the position vector of the target be denoted as $X_{\mathrm{target}}=\left[x_{2},h_{2}\right]$. Then, the relative distance *D* and heading angle ψ can be defined as follows:(14)$$\begin{array}{c} \begin{array}{cc} \; & D=\sqrt{{\left(x_{1}-x_{2}\right)}^{2}+{\left(h_{1}-h_{2}\right)}^{2}},\\ \; & \psi =\arctan\frac{h_{1}-h_{2}}{x_{1}-x_{2}}. \end{array} \end{array}$$

To ensure convergence, the process reward $R_{m}$ is designed in the exponential function form to guide the aircraft toward the target trajectory:(15)$$\begin{array}{c} R_{m}=0.8e^{-5\times 10^{-4}D}+0.2e^{-2\times 10^{-3}\psi }. \end{array}$$

For this flight mission, the overall reward function $R_{\mathrm{total}}$ is a synthesis of the various rewards mentioned above. It ensures that the variant aircraft consistently satisfies process constraints while completing smooth flight during the autonomous decision-making process. The form of the overall reward function $R_{\mathrm{total}}$ is
(16)$$\begin{array}{c} R_{\mathrm{total}}=R_{f}+R_{a}+R_{m}. \end{array}$$

## 4. Morphing Aircraft Tracking Control Method Based on IDDPG

The morphing aircraft, during its flight, not only needs to track the reference trajectory but also requires obstacle avoidance. Thus, we integrate LSTM [19], a network with unique memory units, into both the Actor and Critic networks to receive environment data with temporal features. This makes the learning model better understand the dynamic changes between aircraft states.

### 4.1. LSTM Recurrent Neural Network

The LSTM network is an improved type of recurrent neural network, consisting mainly of four components: the forget gate, input gate, output gate, and memory cell. LSTM not only addresses the short-term memory issue but also mitigates problems such as gradient vanishing and exploding in the loss function. The algorithmic framework is illustrated in Figure 3, where $x_{t}$ represents the input, σ and tanh denote the sigmoid function and hyperbolic tangent function. $f_{t}$, $i_{t}$, and $o_{t}$ express the computed results of the forget gate, input gate, and output gate, respectively. $c_{t}^{\prime }$ represents the candidate for the current memory cell, $c_{t}$ is the updated cell state, and $h_{t}$ is the state of the hidden layer. $w_{f}$, $w_{i}$, $w_{c}$, and $w_{o}$ indicate the weight matrices for the respective components. Thus, $c_{t}$ and $h_{t}$ are expressed as follows:(17)$$\begin{array}{c} \left\{\begin{array}{c} c_{t}=f_{t}^{*}c_{t-1}+i_{t}^{*}c_{t}^{\prime }\\ h_{t}=o_{t}\tanh\left(c_{t}\right) \end{array}\right. \end{array}$$
where
(18)$$\begin{array}{cc} \; & f_{t}=\sigma \left(w_{f}\cdot \left[h_{t-1},x_{t}\right]+b_{f}\right)\\ \; & i_{t}=\sigma \left(w_{i}\cdot \left[h_{t-1},x_{t}\right]+b_{i}\right)\\ \; & c_{t}^{\prime }=\tanh\left(w_{c}\cdot \left[h_{t-1},x_{t}\right]+b_{c}\right)\\ \; & o_{t}=\sigma \left(w_{o}\cdot \left[h_{t-1},x_{t}\right]+b_{o}\right) \end{array}$$

### 4.2. The IDDPG Algorithm Design Process

For complex environments and tasks, learning the optimal policy from scratch after specifying a reward function involves a computationally intensive process. Therefore, a pre-training approach [23] is employed, utilizing expert data to facilitate faster convergence during training.

Specifically, the IDDPG algorithm is based on the incremental kinematic model of a morphing aircraft, dividing the entire reinforcement learning process into two phases: pre-training and DDPG training. Pre-training essentially involves supervised learning, utilizing high-quality data for behavioral cloning to construct the initial action policy. The learning objective expression is
(19)$$\begin{array}{c} L\left(a^{*},\pi_{\theta }\left(s\right)\right)={∥a^{*}-a_{\theta }∥}^{2} \end{array}$$
where $\pi_{\theta }\left(s\right)$ represents the reinforcement learning policy, θ is the hyperparameter for reinforcement learning, $a^{*}$ expresses the expert action, and $a_{\theta }$ is the action output by the learning network. L indicates the error loss function. The closer the output action is to the expert action based on the loss function, the better the learning outcome.

The initial policy network parameters obtained from pre-training are input into the Actor network in the second phase for online training. This stage employs the DDPG algorithm, and as the actions at each time step are deterministic, noise is introduced to enhance policy exploration.
(20)$$\begin{array}{c} a_{t}=\mu \left(s_{t}|\theta^{\mu }\right)+N_{t}, \end{array}$$
where $s_{t}$ and $a_{t}$ represent the state and action at time *t*, respectively. $\mu \left(s_{t}|\theta^{\mu }\right)$ signifies the parameterized policy network responsible for obtaining actions corresponding to the state $s_{t}$. $N_{t}$ denotes time-dependent Ornstein–Uhlenbeck (OU) noise. OU noise differs from Gaussian noise in that the difference between adjacent steps is not significantly large; instead, it tends to explore a certain distance in the direction of the mean based on inertia from the previous step, either positively or negatively. This is advantageous for exploration in a specific direction.

For the studied FWVS aircraft’s flight states in this paper, estimation is performed using a differentiable function approximator $Q\left(s_{t},a_{t}|\theta^{Q}\right)$. At each time step *t*, *N* mini-batch data are sampled from the buffer to update the parameters of the Actor and Critic networks. The parameter updates for the *Q* network employ the temporal-difference algorithm, minimizing the mean squared error of the loss function J at each time step *t*.
(21)$$\begin{array}{c} J=\frac{1}{N}\sum_{t}{\left(y_{t}-Q\left(s_{t},a_{t}|\theta^{Q}\right)\right)}^{2}. \end{array}$$

In reinforcement learning, the objective is to find the optimal policy $\pi^{*}$ that maximizes the expected return. In the case of continuous control, the gradient $\nabla_{\theta^{\mu }}J$ is taken with respect to the hyperparameters θ to update the policy network. By using the expression in Equation (Equation 21), the expression for $\nabla_{\theta^{\mu }}J$ is obtained as shown in Equation (Equation 22).
(22)$$\begin{array}{c} \nabla_{\theta^{\mu }}J\approx \frac{1}{N}\sum_{t}\nabla_{a}Q\left(s_{t},a_{t}|\theta^{Q}\right){|}_{s=s_{t},a=\mu \left(s_{t}|\theta^{\mu }\right)}\nabla_{\theta^{\mu }}\mu \left(s_{t}|\theta^{\mu }\right){|}_{s_{t}}. \end{array}$$

The IDDPG algorithm adopts the framework of DQN with dual networks. It requires simultaneous updating of the hyperparameters for both the *Q* network $Q\left(s,a|\theta^{Q}\right)$ and policy network $\mu \left(s|\theta^{\mu }\right)$. This is typically achieved through the following soft update method:(23)$$\begin{array}{c} \left\{\begin{array}{c} \theta^{Q^{\prime }}\leftarrow \tau \theta^{Q}+\left(1-\tau \right)\theta^{Q^{\prime }}\\ \theta^{\mu^{\prime }}\leftarrow \tau \theta^{\mu }+\left(1-\tau \right)\theta^{\mu^{\prime }} \end{array}\right. \end{array}$$
where $\theta^{Q^{\prime }}$ and $\theta^{\mu^{\prime }}$ represent the corresponding network parameters after updating the *Q* network and the policy network, respectively. τ denotes the soft update rate, with values in the range of [0, 1]. Figure 4 illustrates the learning training framework of the IDDPG policy.

In Figure 4, the environmental model within the entire algorithm framework is the previously established FWVS aircraft. The aircraft, based on the policy network, adjusts the variable-sweep angles, resulting in new flight states. These states are fed back into the policy network, creating a closed loop. Considering the need for the morphing aircraft to choose deformation strategies based on different flight states during training, OU noise is introduced to enhance exploration. The action space is defined as the increments in the variable rates of wings $\delta_{\lambda_{1}}$ and $\delta_{\lambda_{2}}$, and the state space is defined as a continuous six-dimensional flight state. Additionally, the IDDPG algorithm incorporates LSTM networks into both the Actor and Critic networks, enhancing memory for reference trajectories and obstacle positions to further improve algorithm performance. Furthermore, the algorithm stores experiences $\left[s_{t},a_{t},r_{t},s_{t+1}\right]$ in a memory cell during the training process (where $r_{t}$ is the single-step reward generated under the action at time *t*). The learning process maps the current state to the optimal action, and based on the received rewards, computes gradients to update the neural network parameters, ultimately obtaining the optimal deformation strategy during the acceleration climb.

### 4.3. Design and Training of the IDDPG Network

#### 4.3.1. Design of the IDDPG Network

The environmental data received by the aircraft exhibit temporal characteristics during task execution. The LSTM network, with its unique memory units, is better suited for capturing the positions of obstacles and reference trajectories [26].

The environmental data received by the aircraft exhibit temporal characteristics during task execution. The LSTM network, with its unique memory units, is better suited for capturing the positions of obstacles and reference trajectories [26]. The overall network framework of the IDDPG algorithm is illustrated in Figure 5. The Actor network comprises two LSTM layers and one fully connected layer (denoted as FC), responsible for generating control signals, specifically the variable wing sweep rate. The Critic network consists of two parts. The first part, denoted as FC1, is a fully connected network responsible for processing outputs from the Actor network. The second part is an LSTM network, mirroring the structure of the Actor network, tasked with handling environmental state information. Finally, the results from these two parts are processed through another fully connected network, FC2, to calculate and output the *Q* value of the Actor network. Considering the task environment, 128 hidden layer neurons were chosen for the LSTM network.

From the above algorithmic process, it is evident that the IDDPG algorithm is based on the Actor-Critic network. The structure of the network significantly influences the algorithm’s performance, necessitating design optimization.

The Actor network is the policy network, established through a neural network to map from the current state to the next action (Figure 6). In the IDDPG algorithm, the input to the Actor network consists of the aircraft’s state variables and altitude error, forming a seven-dimensional array $s_{t}=\left[x,h,V,\gamma ,\alpha ,q,\Delta h\right]$. This network consists of two hidden layers, with the first layer being an LSTM network housing 128 hidden neurons to effectively process environmental information, enhancing real-time performance [27]. The second layer employs fully connected architecture with 256 hidden neurons. Additionally, the rectified linear unit (Relu) function is chosen as the activation function between each hidden layer to reduce the probability of gradient vanishing issues [28]. Data from the first two layers flow directly into the subsequent layer through the activation function, and the final layer connects to the two-dimensional action space. Considering that the actions (sweep angle deformation rate) are constrained, the Tanh activation function is chosen for the final layer to limit the output within a specified range.

In the IDDPG algorithm, the Critic network establishes a mapping from actions and states to the *Q* function through a neural network, with the structure designed as depicted in Figure 7. The input layer of the Critic network is composed of state variables and the output actions from the Actor network, forming a nine-dimensional array. The output represents the *Q*-function, denoted as $Q\left(s,a\right)$. There are two fully connected hidden layers, with the first and second layers having 128 and 200 hidden neurons, respectively. Unlike the Actor network, the final layer is directly connected to a one-dimensional output layer, which is further used to iteratively update the Bellman equation in reinforcement learning to maximize the output *Q* value.

#### 4.3.2. IDDPG Network Parameter Settings

In the IDDPG algorithm, apart from the Actor–Critic network structure, another crucial factor influencing algorithm performance is the setting of hyperparameters. This includes the total number of steps, total number of episodes, Actor learning rate, Critic learning rate, noise variance, and discount factor, among others. Based on empirical findings in the references and extensive trial-and-error simulations, the final values and definitions of all hyperparameters are presented in Table 1.

It should be noted that the above parameter settings were obtained through extensive trial and error.

Here, an episode is defined as the process in which the agent executes a particular policy in the environment from start to finish. The IDDPG algorithm defaults to a maximum of 55,000 episodes, with the first 20,000 episodes constituting an offline pre-training process using expert data to avoid inefficient random exploration in the early stages of the IDDPG algorithm. Due to a sampling time of 0.02 s, the maximum number of steps per episode is set to 800, ensuring that the total time for each episode does not exceed 16 s. The Actor learning rate and Critic learning rate define the step size for gradient updates during network parameter optimization. After multiple trials, the Actor and Critic learning rates are determined to be 0.0001 and 0.001, respectively. The buffer size for storing past experiences is set to 1,000,000, and the batch size of samples used during each gradient update is determined to be 256.

Another innovation of the IDDPG algorithm is the introduction of noise to increase sample diversity. In this study, OU noise with a variance of 0.15 is employed. Finally, considering that the initial actions may not necessarily lead to the expected state, the weights for future rewards need to decrease gradually. Therefore, a discount factor is introduced to calculate the expected future returns during the learning process and adjust the impact of future rewards, typically chosen as 0.991.

## 5. Simulation and Analysis

### 5.1. IDDPG Algorithm Network Training

To validate the proposed IDDPG algorithm framework, simulations are conducted using the reinforcement learning toolbox in Matlab. Initially, without incorporating the kinematic incremental model, the impact of integrating LSTM networks into the DDPG algorithm will be assessed. The learning environments for both DDPG and LSTM–DDPG are constructed using the nonlinear expression (Equation 4) of the FWVS aircraft model, along with the state and action spaces, and reward function proposed in Section 3.2. However, the non-sparse reward state is replaced with the following form:(24)$$\begin{array}{c} R_{m}^{\prime }=w_{4}r_{h}+w_{5}r_{V}+w_{6}r_{q}, \end{array}$$
where $r_{h}=e^{-5\times 10^{-4}\sqrt{\left(h-h_{\mathrm{target}}\right)}}$ represents altitude reward, $r_{V}=e^{-2\times 10^{-3}\sqrt{\left(V-V_{\mathrm{target}}\right)}}$ is flight speed reward, and $r_{q}=e^{-5\sqrt{\left(q-q_{\mathrm{target}}\right)}}$ denotes pitch angle rate reward. The weights $w_{i}$ assigned to each reward are designed as $\left[0.5,0.4,0.1\right]$. When both velocity and altitude reach the target endpoint, the corresponding rewards reach their maximum values, and the weighted state reward reaches 1. Table 2 displays the reward function parameters determined after multiple simulation trials.

The network training performance of the LSTM–DDPG algorithm is compared with the traditional DDPG algorithm [29], and the results are illustrated in Figure 8. It is evident that the LSTM–DDPG algorithm converges more rapidly to a higher average reward and exhibits greater stability. The incorporation of LSTM networks enables the DDPG algorithm to better discern valuable data, leading to optimal results and avoiding unnecessary exploration during the climb phase of the FWVS aircraft. This highlights the excellent performance of the LSTM network.

Subsequently, the IDDPG algorithm, generated based on the kinematic increment model incorporating a reference trajectory, is employed to validate whether the inclusion of trajectory constraints accelerates the convergence speed of the network. In this simulation, the reinforcement learning environment utilizes the incremental model (Equation 7) of the morphing aircraft, with an action space $a=\left[\delta_{\lambda_{1}},\delta_{\lambda_{2}}\right]\in \left[-0.1,0.1\right]$ and hyperparameters referenced in Table 2. Furthermore, in comparison to LSTM–DDPG, the IDDPG algorithm incorporates the state reward function $R_{m}$, as depicted in Equation (Equation 15), within the reward function module. The simulation results are illustrated in Figure 9. Table 3 presents the average single-training time, total training time, and average reward for three algorithms: DDPG [29], LSTM–DDPG, and IDDPG.

The simulation platform of each algorithm in Table 3 is the Mac system M2 chip, and the Matlab version is 2023b.

Based on the simulated training process and Table 3, it can be observed that IDDPG significantly reduces training time compared to DDPG, while achieving substantial improvements in training effectiveness and reward values. Additionally, as depicted in Figure 9, the LSTM–DDPG algorithm, without environmental modifications, achieves successful episodes around 8000, with the average reward converging after approximately 15,000 episodes. In contrast, the IDDPG algorithm, after redesigning the action space, achieves successful episodes before 5000 and starts converging to the maximum average reward just beyond 10,000 episodes. This indicates that the incremental model and action space derived from the reference trajectory can reduce the exploration space for actions, thereby accelerating the convergence rate (Table 3).

### 5.2. Flight Control Simulation Based on IDDPG Algorithm

Upon completion of the training for the FWVS aircraft control strategy, simulation testing is conducted. The testing set is configured for 1000 episodes to assess the deformation effectiveness and success rate of accomplishing the tasks for the FWVS aircraft. The initial state is set to $s_{0}=\left[0\;m,200\;m,20\;m/s,0\;\mathrm{rad},0.0799\;\mathrm{rad},0\;\mathrm{rad}/s\right]$, and the target states, represented by $h_{f}=232\;m$, $V_{f}=32\;m/s$, and $q_{f}=0\;\mathrm{rad}/s$ are defined. The objective is to complete the process of accelerating climb within a total time not exceeding 16 s. To evaluate the algorithm’s adaptability to different initial environments, random initial heights are selected:(25)$$\begin{array}{c} h_{0}=200+rand\left(0.5\right), \end{array}$$
where $rand\left(\zeta \right)$ represents a random number within the range of ζ. In this simulation, the uncertain parameter ε in Equation (Equation 8) is set to 0.1, indicating a 10% deviation in the lift coefficient under the current conditions. To validate the proposed algorithm’s effectiveness compared to the traditional DDPG method [29], in this example, we trained and simulated both the proposed IDDPG and the DDPG methods separately.

After validation, employing the agent trained through IDDPG as the controller for the aircraft resulted in a success rate of 91.32%, whereas DDPG achieved a success rate of 78.26%. For different initial altitudes, Figure 10 illustrate the state variations for the FWVS aircraft under the training agents of IDDPG and DDPG.

It should be noted that, once the learning network is trained, it can rapidly output action commands based on the current state of FWVS. The time required for the trained learning network to output action commands based on the current state of FWVS is on the order of $10^{-2}$ seconds (this is the result of running on the Mac M2 chip, Matlab2023b).

From Figure 10, it can be observed that, under the DDPG−trained agent, the FWVS aircraft (all black curves) deviates from the target point under different initial altitudes and lift coefficient uncertainties. In contrast, the FWVS aircraft under the influence of the IDDPG−trained agent (all red curves) reaches the target point despite slight oscillations compared to the corresponding DDPG curves. Furthermore, all state variables throughout the entire flight process satisfy the imposed constraints, successfully accomplishing the task of ascending to level flight.

Figure 11 illustrates the control variations of IDDPG for completing the flight task. It is observed that, during the climbing phase, the primary adjustments are made to the trailing−edge wing, while in the acceleration phase, the leading−edge wing plays a predominant role after reaching a certain altitude. However, the results indicate noticeable oscillations in control inputs, suggesting a potential issue with a large action space. Figure 12 visually presents the aircraft shape during the flight process. In the climbing phase, the trailing−edge sweep angle gradually increases, generating a pitching moment to increase lift. After reaching the target altitude, the trailing−edge sweep angle reaches its maximum value. At this point, the leading edge sweep angle is employed to control the pitch−down moment, maintaining a level flight state.

Furthermore, employing the trained network, simulation experiments are conducted for trajectory tracking control and obstacle avoidance control. A test set of 1000 episodes is utilized to evaluate the effectiveness of tracking a reference trajectory. The various initial values, distributed on either side of the reference trajectory, are chosen to assess the tracking performance. Additionally, an uncertainty coefficient ε for lift is set to 0.1. The reference trajectory is depicted by the solid blue line in Figure 13.

As depicted in Figure 13, post−training with the IDDPG algorithm, the FWVS aircraft demonstrates rapid and accurate trajectory tracking to accomplish the task of accelerated climb, even in the presence of initial state errors and lift coefficient uncertainty. This validates the robustness of the IDDPG strategy to uncertainties in the deformation process.

Moreover, due to the unique memory units of the LSTM network, the UAV can better record obstacle positions and consider information from previous time steps when selecting actions. To further validate the information processing capabilities of the IDDPG network, obstacles are introduced into the flight environment to create different scenarios. A comparison is then made with the traditional DDPG method to assess its performance against the proposed approach. The initial state is set to $s_{0}=\left[0\;m,200\;m,20\;m/s,0\;\mathrm{rad},0.0799\;\mathrm{rad},0\;\mathrm{rad}/s\right]$, with the reference trajectory represented by the solid orange line in Figure 14. The obstacle is positioned at x=150m,h=215m, and the circular obstacle has a radius of 1 m, as depicted by the black circle in Figure 14. Trajectory test for obstacle avoidance is conducted with the IDDPG network, achieving a success rate of at least 78.5% after numerous tests, whereas DDPG achieved only 48% success in a comparable scenario. Figure 14 illustrates the flight trajectories for obstacle avoidance during the 500th, 600th, and 899th instances.

From Figure 14a, it can be observed that the FWVS aircraft trained with the IDDPG algorithm successfully resumes tracking the reference trajectory after avoiding obstacles. However, under the DDPG−trained agent (Figure 14b), the FWVS aircraft deviates from the target trajectory while avoiding obstacles and struggles to reach the target point. Here, “*i*” denotes the number of successful episodes.

## 6. Conclusions

This paper focuses on the trajectory tracking control of a FWVS aircraft using reinforcement learning, proposing the IDDPG algorithm. This algorithm not only utilizes LOS for target point tracking, enhancing the algorithm’s realization rate in target point tracking, but also integrates LSTM units to improve the algorithm’s adaptability to environmental changes during flight. Subsequently, through a significant amount of training, it is concluded that the IDDPG algorithm has a faster convergence speed than the DDPG algorithm, and it also achieves a good tracking control effect for the FWVS with uncertainties. Lastly, to further validate the adaptability of this controller to environmental disturbances, simulations are conducted by introducing environmental perturbations. The results indicate that the FWVS aircraft can successfully navigate around obstacles and retrace the reference trajectory. 

## Figures and Tables

**Figure 1 biomimetics-09-00263-f001:**
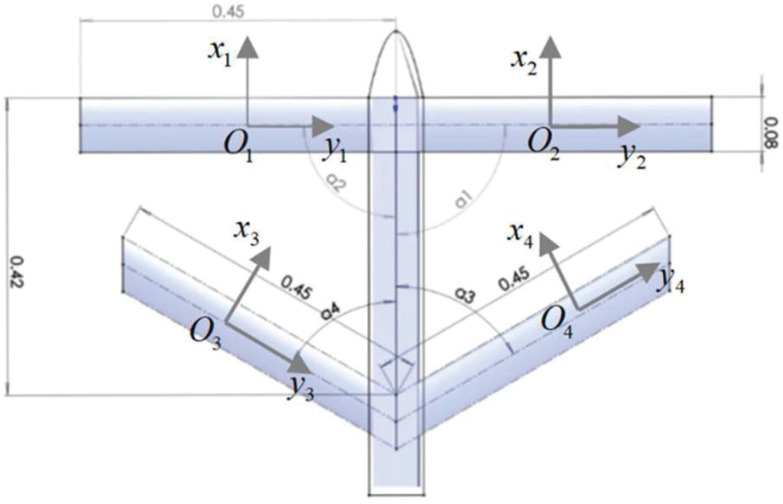
Schematic diagram of the FWVS aircraft.

**Figure 2 biomimetics-09-00263-f002:**
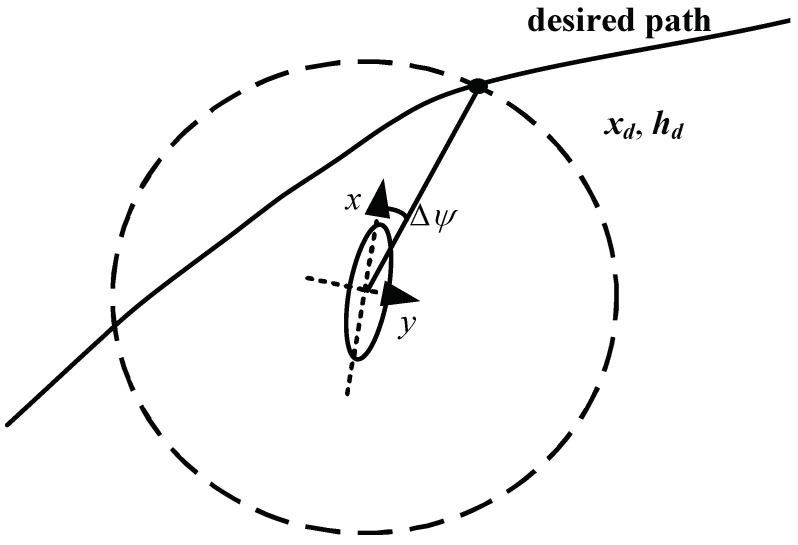
Schematic diagram of LoS path tracking method.

**Figure 3 biomimetics-09-00263-f003:**
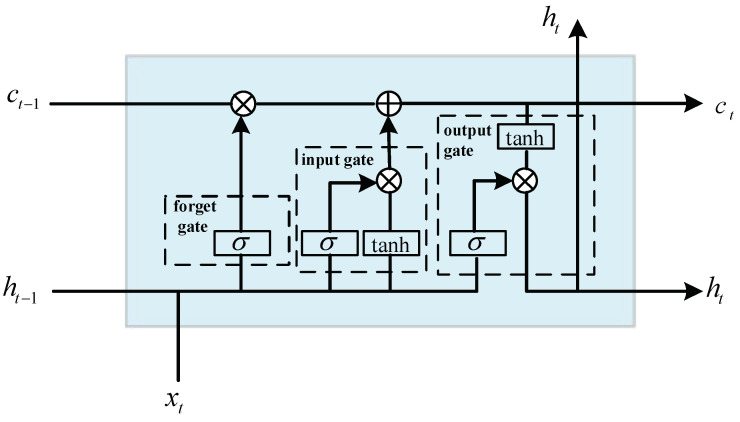
LSTM algorithm framework.

**Figure 4 biomimetics-09-00263-f004:**
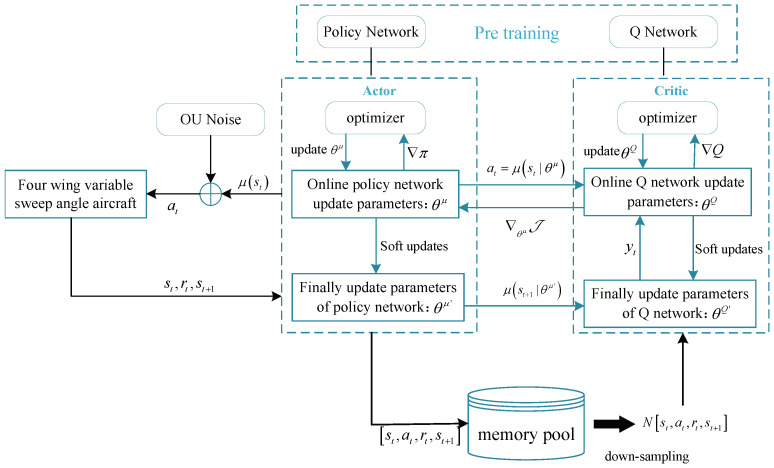
Learning and training framework of FWVS aircraft.

**Figure 5 biomimetics-09-00263-f005:**
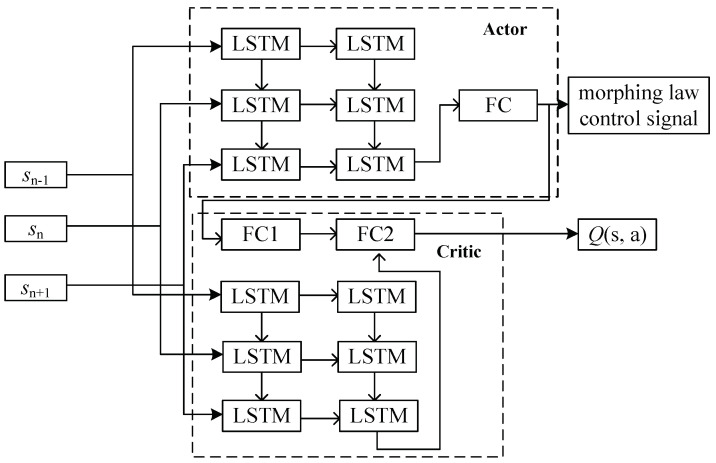
LFramework of the IDDPG network.

**Figure 6 biomimetics-09-00263-f006:**
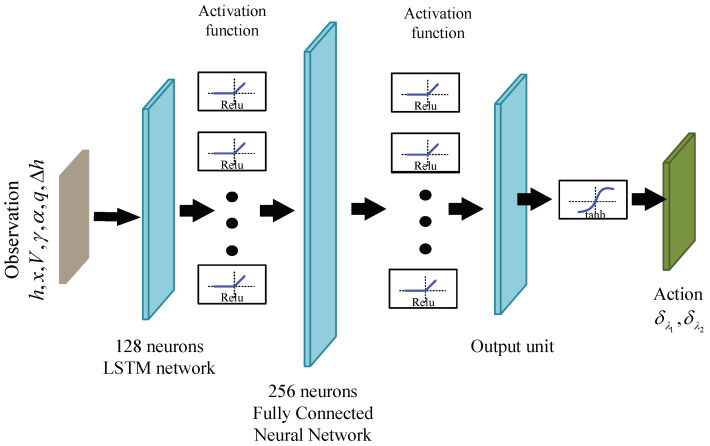
Actor network structure diagram.

**Figure 7 biomimetics-09-00263-f007:**
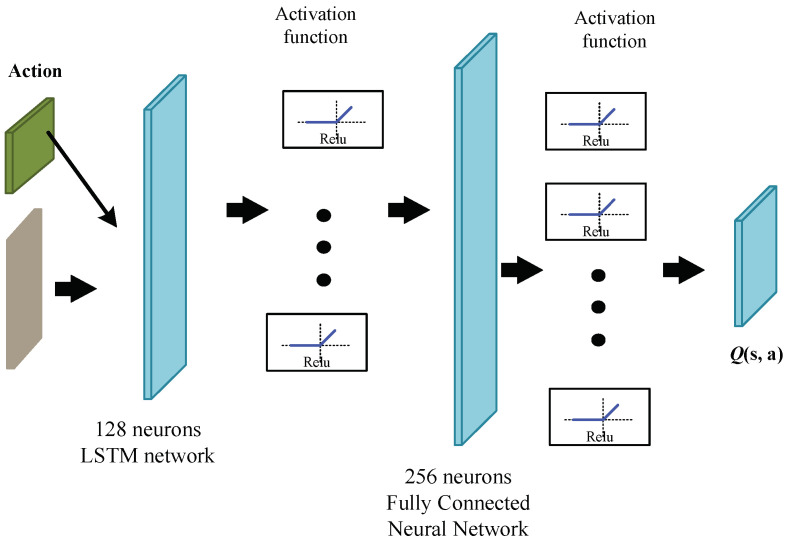
Critic network structure diagram.

**Figure 8 biomimetics-09-00263-f008:**
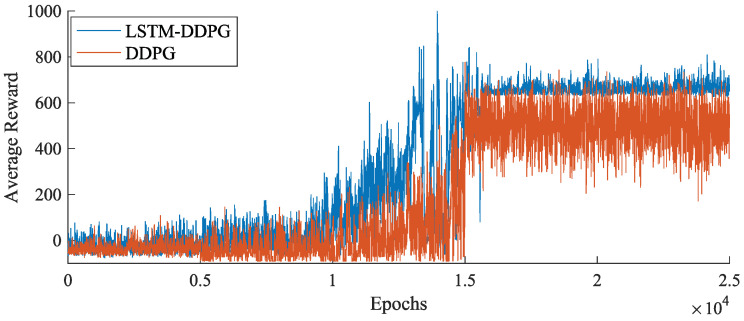
Comparison of average rewards for LSTM–DDPG and DDPG algorithm training.

**Figure 9 biomimetics-09-00263-f009:**
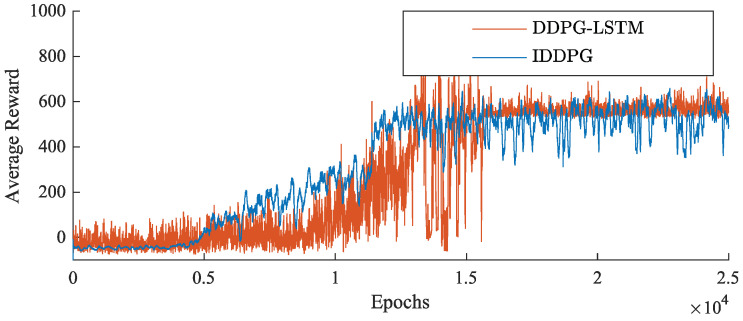
Algorithm training average reward comparison chart.

**Figure 10 biomimetics-09-00263-f010:**
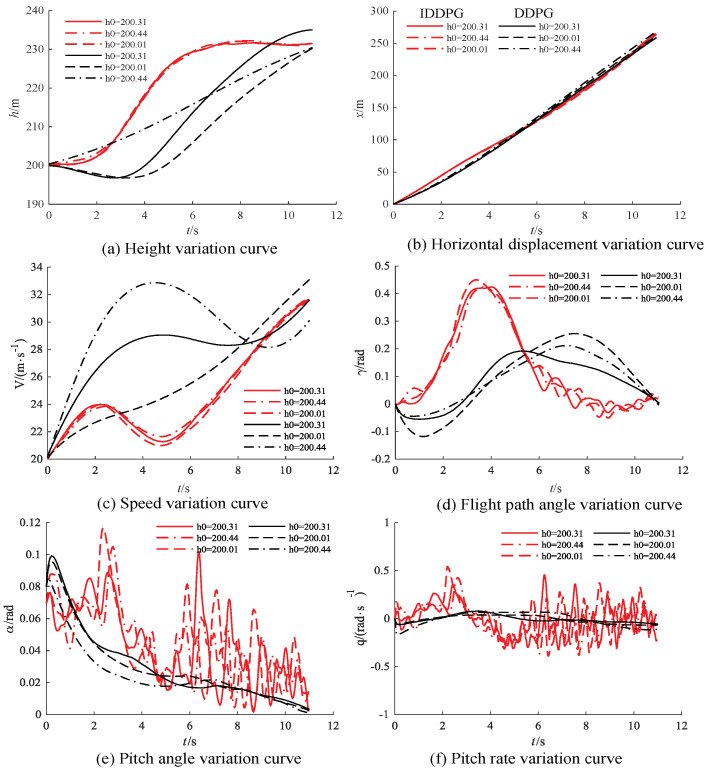
State variable variation curves under different initial states (the red curve is IDDPG, the black curve is DDPG).

**Figure 11 biomimetics-09-00263-f011:**
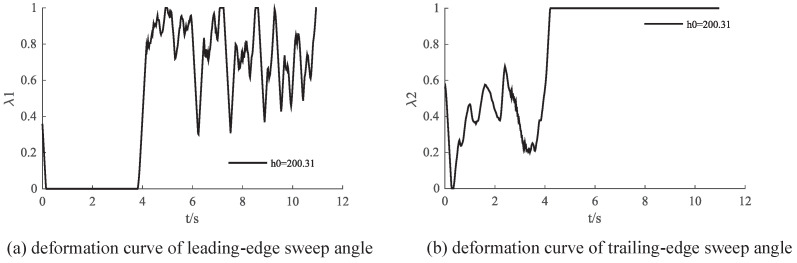
Action variation curve.

**Figure 12 biomimetics-09-00263-f012:**
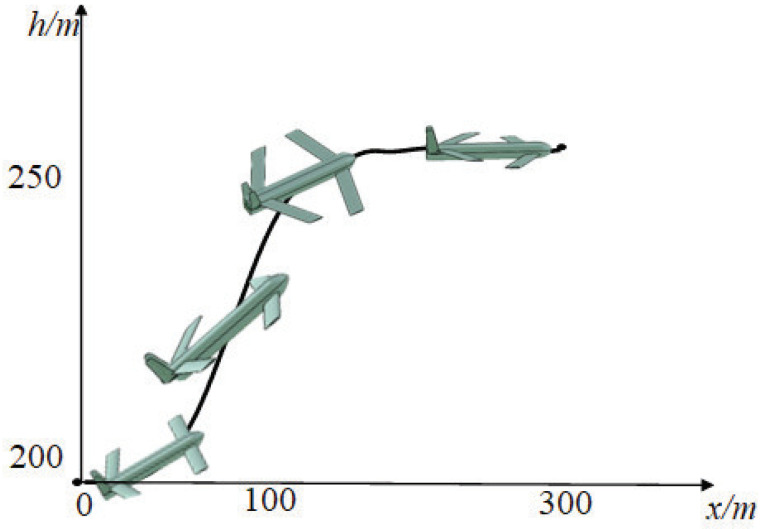
Outline diagram of the aircraft during flight.

**Figure 13 biomimetics-09-00263-f013:**
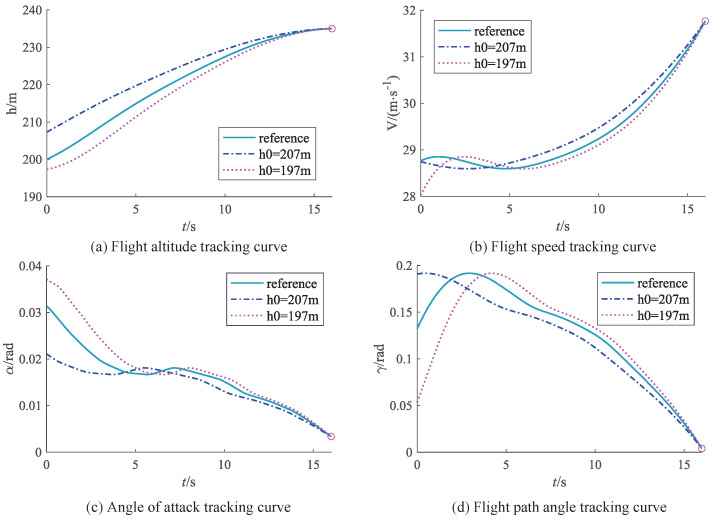
Tracking curves of reinforcement learning controllers under different initial values.

**Figure 14 biomimetics-09-00263-f014:**
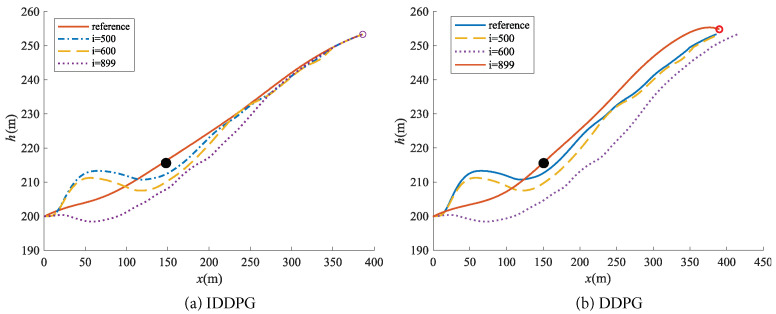
Tracking path of morphing aircraft with different iterations (The black dot is the obstacle added during algorithm training).

**Table 1 biomimetics-09-00263-t001:** The hyperparameter of IDDPG algorithm.

Hyperparameter	Parameter Value
Target network update rate	0.001
Actor network learning rate	0.0001
Critic network learning rate	0.001
Experience replay pool capacity	$10^{6}$
Number of small batch samples	256
Discount factor	0.991
Maximum number of episodes	80,000
Maximum steps per episode	800
Sampling time (s)	0.02

**Table 2 biomimetics-09-00263-t002:** Reward function parameters.

Reward Function Parameters	Parameter Value
$h_{\mathrm{target}}$	232 m
$V_{\mathrm{target}}$	32 m/s
$q_{\mathrm{target}}$	0 rad/s
$\delta_{h}$	1 m
$\delta_{V}$	0.5 m/s
$\delta_{\gamma }$	0.1 rad
$\delta_{q}$	0.1 rad/s
$h_{\max},h_{\min}$	235 m, 198 m
$V_{\max},V_{\min}$	33 m/s, 20 m/s
$\alpha_{\max}$	1 rad/s
$q_{\max}$	ππ44 rad

**Table 3 biomimetics-09-00263-t003:** Algorithm comparison.

Algorithms	DDPG [29]	LSTM–DDPG	IDDPG
Single-training time	0.31 s	0.35 s	0.35 s
Total training time	37.5 h	30.8 h	18.2 h
Average reward	531	787	791

## Data Availability

Data are contained within the article.

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
