# Peer review of "Trajectory Tracking Control of Variable Sweep Aircraft Based on Reinforcement Learning"

_biomimetics, 2024, doi:10.3390/biomimetics9050263_

Round 1

Reviewer 1 Report (New Reviewer)

Comments and Suggestions for Authors

The authors mentioned in the conclusion that LSTM network is integrated into the DDPG algorithm framework, forming the IDDPG algorithm which not exactly true. In the algorithm comparison results of Table 3, LSTM-DDPG and IDDPG both yield different total training time and average reward.

In lines 140-141, it was given a 20% uncertainty in the lift coefficient. In lines 443-444, the uncertainty coefficient for lift is chosen to be 0.1 ?

Author Response

Dear reviewer, the reply to your comments has been placed in the attachment
'Response letter_Biomimetics To Reviewer 1', please check.

Reviewer 2 Report (New Reviewer)

Comments and Suggestions for Authors

The paper presents an interesting RL approach combined with elements of deep learning for efficient variable sweep aircraft control. There are a few important drawbacks in this study, specifically:

1) DQN is referred to as static, i.e., not taking into account current flight characteristics (state); however, LSTM is used here with a "reference trajectory", which is marginally different in principle.

2) In section 3.1, the "standard Markov" approach is unclear, specifically regarding the exact discretization method and how this affects the overall predictive quality and granularity of the proposed model.

3) In section 3.2, using the "reference trajectory" or "desired path", the specific delta (error step) is purely linear in the "sparse" component and almost as such in the "dense" component (e.g. see Eq.9, Eq.19, Eq.24, etc). The authors clearly identify this task as such, referring to "trajectory tracking" (see pg.17). Therefore, there is no clear advantage of why apply an RL approach when the same task can be accomplished by standard PID controllers and/or Kalman filtering (e.g. see Fig.2). At the very least, these should be the comparison baseline in the experimental protocol against the proposed approach.

4) In the model description there are many ad-hoc coefficients for approximators and learning (e.g. see Table 1-2 and section 4.3.2 descriptions); these need to be clarified w.r.t. how they were fixed and what is the effect of changing them or re-estimating them in a different model setup.

5) In section 5.1, training times are mixed with description/discussion about the model (methodology); furthermore, not specific hardware platform is described as part of a clear experimental protocol.

6) In section 5.2, the evaluation upon simulations is referred to as "accuracy rate"; this is invalid, since this is not a typical classification task and, thus, no such performance metric can be obtained. This is highly problematic w.r.t. the experimental setup and performance evaluation.

7) In Fig.14(a) the "correct" reference trajectory is clearly colliding with the noted obstacle; the simulation tests and the experimental protocol in general seems extremely problematic. Also, there is no clear presentation of results, statistical significance, etc (e.g. see: "success rate of at least 78.5% after a single test").

8) The general question is: If training time of this proposed approach is in the order of tens of hours, then what is the gain of it over a very analytical kinematic model? How is this compatible with the need to implement and embed is in real-time FWVS control?

9) There are multiple sections of text highlighted (see pg.14-16) - not sure if it intentional or not by the authors, but clearly these need to be removed; also, careful review of the text is needed for typos and other errors.

Comments on the Quality of English Language

(see comment 9 above)

Author Response

Dear reviewer, the reply to your comments has been placed in the attachment
'Response letter_Biomimetics To Reviewer 2', please check.

This manuscript is a resubmission of an earlier submission. The following is a list of the peer review reports and author responses from that submission.

Round 1

Reviewer 1 Report

Comments and Suggestions for Authors

The reviewed article relates thematically to UAV control algorithms and presents the IDDPG algorithm, based on artificial intelligence methods.

The structure of the article is correct, the literature has been selected correctly, but in its current form the paper needs some improvements.

1) There is a lack of a solid literature review to justify the research and to set the content in a certain framework - the bibliography counts only 20 items.

2. While the supposed effects of the method are presented, a sound comparison with the other methods is missing, especially in the context of results related to the efficiency.

3. The discussion and conclusion do not convince the reader of the justified need for research towards the development of a new method - here again, it is necessary to refer to results from the current literature and compare them with those obtained by the Authors. Only on this basis will it be possible to conclude on the contribution made to the field.

Reviewer 2 Report

Comments and Suggestions for Authors

the authors present and analyse a very contemporary problem. In general, the presentation of the idea that led to this problem is very good. They analyze and describe the problem satisfactorily and at the end they present some results. I think that this article is suitable for publication